# Effects of Oligofructose Supplementation on Growth Performance, Antioxidant Capacity, Immunity, and Intestinal Health in Growing Rabbits

**DOI:** 10.3390/ijms26178694

**Published:** 2025-09-06

**Authors:** Liwen Qin, Chunlong Xiao, Menglei Shi, Lu He, Yifei Du, Lifan Lin, Zekai Zhang, Yichen Lin, Yue Feng, Qinghua Liu, Changchuan Ye

**Affiliations:** Department of Animal Science, College of Animal Science, Fujian Agriculture and Forestry University, Fuzhou 350002, China; 18706011090@163.com (L.Q.); 13799729165@163.com (C.X.); smlei77@163.com (M.S.); luheluhe0202@163.com (L.H.); dyfrjl@163.com (Y.D.); lifan_lin2003@163.com (L.L.); 15985874725@163.com (Z.Z.); a15960022520@163.com (Y.L.); yuefeng@fafu.edu.cn (Y.F.)

**Keywords:** oligofructose, rabbits, antioxidant capacity, immunity, intestinal health

## Abstract

This study aims to investigate the effects of dietary supplementation with different concentrations of oligofructose (FOS) on growth performance, antioxidant capacity, immunity, and intestinal microbial composition in growing rabbits. One hundred female Dehua black rabbits (34 d of age) were randomly assigned to four groups (CON, FOS-1, FOS-2, and FOS-3), with twenty-five rabbits in each group. The CON group received only a basal diet, while the FOS-1/FOS-2/FOS-3 group received the diet supplemented with 0.3%/0.6%/0.9% FOS, respectively. The trial period lasted for 72 days. Our results revealed that FOS supplementation could improve the growing performance of rabbits and decrease the feed/gain ratio. FOS significantly enhanced serum antioxidant enzyme (SOD) and total antioxidant capacity (T-AOC) while reducing malondialdehyde (MDA). The levels of plasmic immunoglobulin (IgG, IgA, and IgM) and intestinal immune factors (IL-1α, IL-2, and sIgA) were significantly improved with the FOS supplement. Additionally, FOS can improve intestinal morphology and enhance the activity of intestinal enzymes, including cellulase, lipase, and protease. Furthermore, FOS supplementation influenced the composition of intestinal microflora by increasing the abundance of *Lachnospiraceae*_NK4A136_group (barrier-enhancing) and *Monoglobus* (fiber-degrading). In conclusion, the addition of FOS has a positive impact on the growth performance, antioxidant capacity, immunity, and intestinal health of growing rabbits. The optimal dietary addition for rabbits was identified as 0.6% oligofructose.

## 1. Introduction

Intestinal diseases constitute one of the primary causes of morbidity and mortality in farming animals such as pigs or rabbits [1]. Consequently, enhancing intestinal health is a crucial means to improve animal survival rates and enhance the efficiency of livestock production. Incomplete digestive tract development and inadequate intestinal immunity render neonatal animals susceptible to oxidative stress and infection by pathogenic bacteria, thereby inducing a series of diseases. Extensive previous research has confirmed that intestinal microbiota indeed assist in and maintain metabolic functions, influence immune system development and function, and enhance resistance to stress and antioxidant capacity [2,3,4]. The gut microbiota extracted from human neonates can promote the development of innate and adaptive immunity in germ-free animals (such as germ-free mice) and alleviate intestinal inflammatory responses [5]. Healthy intestinal microbiota can assist the host in resisting pathogenic bacteria, modulating the intestinal immune system and enhancing the antioxidant capacity of the host. Early-life intervention in animals aids in the rapid establishment of a beneficial gut microbiota in neonates, which could promote the subsequent growth, immune system development, and healthy development of the animals [6].

Prebiotics can elicit specific changes in the composition and function of the gut microbiota, exerting beneficial effects on the host [7]. Most prebiotics belong to the category of non-starch polysaccharides, including oligofructose (FOS), xylan, and pectin. Prebiotics can stimulate the proliferation and metabolic activities of beneficial microorganisms and promote the proliferation and differentiation of intestinal cells, thereby contributing to the establishment of a stable microbial ecosystem [4]. A study has indicated that feeding inulin to sows during gestation and lactation can increase the abundance of *Enterococci* in the feces of both sows and their offspring [8]. By promoting the growth of beneficial intestinal bacteria, prebiotics can effectively inhibit the colonization and multiplication of pathogenic bacteria [9,10,11,12], thus favoring intestinal health.

FOS can regulate intestinal microecology, improve immunity, and promote digestion and absorption [13,14,15,16]. Since the digestive enzymes of animals cannot degrade oligofructose molecules bound by β-1,2 glycosidic bonds, FOS will enter the posterior intestinal tract in an intact form to be utilized by the beneficial intestinal bacteria [17]. Notably, different doses of oligofructose exhibit varying effects on farmed animals. Alimentary FOS overload will reduce antioxidant levels conversely and induce acute laminitis in dairy heifers [18]. Therefore, systematic investigations are imperative to determine the optimal additive dosage that harmonizes production efficacy with metabolic safety.

In this study, we investigated the effects of various levels of FOS in the diets of rabbits. The FOS was extracted from a kind of *Poaceae* family plant (*Pennisetum giganteum*). We focused on its impact on growth performance, immunity, antioxidant capacity, and intestinal health. Our study aims to find out the potential growth-promoting mechanism of FOS. Additionally, our findings aim to explore the correlation between antioxidant capacity and intestinal health and thus provide valuable references for further studies and applications of FOS.

## 2. Results

### 2.1. Growth Performance and Meat Quality

As can be seen from Table 1, the dietary supplementation with 0.6% FOS (FOS-2 group) showed the highest final body weight (*p* < 0.05). FOS increased (*p* ≤ 0.05) the average daily gain (ADG) and feed/gain ratio (F/G) compared to the control. Additionally, 0.6% and 0.9% FOS highly significantly improved (*p* < 0.01) cold carcass weight compared to the control. As for meat quality, FOS significantly decreased (*p* < 0.05) the drip loss rate and shear force of the dorsal and leg muscle compared to the control. In particular, the pH_45min_ value of the dorsal muscle in the FOS-2 group (0.6% FOS) was significantly higher than that in the control group (*p* < 0.05).

### 2.2. Changes in Plasmic Metabolites and Antioxidant Capacity

As shown in Table 2, the FOS supplement in the diet significantly reduced the level of total cholesterol (TC) (*p* < 0.01) in the plasma. Similarly, triglyceride (TG) levels were significantly decreased in the FOS-2 and FOS-3 groups (*p* < 0.01). High-density lipoprotein cholesterol (HDL-C) levels were significantly reduced in both the FOS-2 and FOS-3 groups following FOS supplementation (*p* < 0.01). Additionally, only the FOS-3 group showed a significant reduction in low-density lipoprotein cholesterol (LDL-C) levels (*p* = 0.03) compared to the CON group. The FOS-2 and FOS-3 groups exhibited significant reductions in FFA (*p* < 0.01). In addition, the total antioxidant capacity (T-AOC) of all the experimental groups (FOS-1, FOS-2, and FOS-3) significantly increased compared with the CON group (*p* < 0.01). The addition of FOS to feed would significantly enhance SOD activity in plasma (*p* < 0.05), while the FOS-2 group showed the highest superoxide dismutase (SOD) activity. Notably, the level of malondialdehyde (MDA) in all the experimental groups significantly decreased compared to the CON group (*p* < 0.01), indicating a positive effect on antioxidant status. The FOS supplement in the diet significantly increased the level of Mg (*p* < 0.01) and Fe (*p* < 0.01) in the serum. In particular, the dietary supplementation with 0.6% FOS (FOS-2 group) significantly elevated the content of Ca (*p* = 0.05).

### 2.3. Changes in Immunological Function

As shown in Figure 1, dietary supplementation with FOS elevated immunoglobulin G (IgG) and immunoglobulin A (IgA) levels in plasma (*p* < 0.01). The levels of immunoglobulin M (IgM) in plasma were significantly higher in the FOS-1 and FOS-2 groups (*p* < 0.01). The dietary supplementation with 0.3% and 0.6% FOS significantly increased the level of interleukin-1α (IL-1α) and secretory immunoglobulin A (sIgA) (*p* < 0.01) in the rabbits’ jejunum. Additionally, the levels of interleukin-2 (IL-2) in the jejunum were also observed to have a significant improvement in the groups with FOS treatment (*p* < 0.01). Significant improvement was observed in the levels of ileal IL-1α in all the experimental groups (*p* < 0.01). For IL-2 and sIgA levels, significant elevations were observed in the FOS-1 and FOS-2 groups in the ileum (*p* < 0.01), compared with the control group.

As shown in Figure 2, dietary supplementation with 0.3% FOS (FOS-1 group) significantly increased the liver index and kidney index (*p* < 0.01). However, there were no significant differences in the thymus index and spleen index among these groups (*p* > 0.05).

### 2.4. Changes in Intestinal Morphology and Intestinal Enzyme Activities

The microscopic images of the hematoxylin–eosin (HE) slide are shown in Figure 3 while the histological analyses are summarized in Table 3. According to Table 3, dietary supplementation with FOS significantly increased villus height only in the ileum (*p* < 0.05). The other two segments did not show a difference in villus height among these groups. Crypt depth in the jejunum significantly decreased in the FOS-2 and FOS-3 groups (*p* < 0.01). However, we found that the villus height/crypt depth ratio (V/C ratio) of the rabbits significantly increased with FOS treatment. In particular, the V/C ratio in the duodenum increased in the FOS-2 group (*p* < 0.05) and the V/C ratio in the jejunum increased in the FOS-3 group (*p* < 0.05). All experimental groups showed a significantly higher V/C ratio in the ileum (*p* < 0.05), compared to the CON group. Compared to the other two experimental groups (FOS-1 and FOS-2 groups), the duodenal villus length was increased (*p* < 0.05) in the group supplemented with 9% FOS (FOS-3 group). Additionally, the ileal villus length in the FOS-3 group was higher compared to the CON group and FOS-2 groups (*p* < 0.05). No significant changes in villus thickness were observed among these groups in the duodenum, ileum, or jejunum. The number of goblet cells in the ileum significantly increased in all FOS treatment groups (*p* < 0.01), with the FOS-3 group exhibiting the highest increase (*p* < 0.01). Meanwhile, dietary supplementation with 6% and 9% FOS (FOS-2 and FOS-3 groups, respectively) showed the higher goblet cell density in the jejunum, compared to the CON group.

As shown in Table 4, the activities of cellulase, lipase, and total protease in the experimental groups were significantly higher than those in the CON group (*p* < 0.01). Specifically, dietary supplementation with 0.6% FOS (FOS-2 group) showed the highest activities of these three intestinal enzymes among these groups (*p* < 0.01). These results demonstrate that the addition of FOS can enhance the activity of digestive enzymes in the intestine, thereby improving digestive capacity for the nutrients of rabbits.

### 2.5. Changes in Short-Chain Fatty Acid Profile in Cecal Contents

As shown in Figure 4, our results indicated that dietary supplementation with 0.9% FOS (FOS-3 group) exhibited a significant reduction in acetic acid content compared to other groups (*p* < 0.05), which suggested that excessive oligofructose might have a potential inhibitory effect on cecal fermentation. No significant differences were found in propionic acid, butyric acid, valeric acid, caproic acid, isobutyric acid, and isovaleric acid among all the groups.

### 2.6. Intestinal Microbiota Community

As illustrated in the Venn diagram (Figure 5A), a total of 1393 OTUs were identified across all four experimental groups. Among these, only 90 OTUs (CON: 23, FOS-1: 17, FOS-2: 20, FOS-3: 30) were unique to each group. This substantial overlap suggests a core microbiome structure conserved across treatments. Dilution curve analysis (Figure 5B) revealed that all curves transitioned from steep inclines to plateaus, indicating that the sequencing depth was sufficient to capture the majority of bacterial diversity present in the cecal samples. The extended horizontal axis span observed in the rank-abundance curves (Figure 5C) demonstrated high species richness in all groups. While vertical axis uniformity across groups suggested comparable evenness distributions, the pronounced vertical gradients indicated significant inter-species abundance disparities within each community.

The alpha-diversity analysis is shown in Table 5; all groups exhibited 99% sequence coverage, confirming that the sequenced libraries effectively represented the bacterial diversity within cecal contents. There were no significant differences in ACE, Chao1, and Shannon indices across all groups (*p* > 0.05).

The beta-diversity analysis is shown in Figure 6. NMDS analysis was performed on bacterial communities based on the Bray–Curtis distance algorithm. The stress value was 0.1370 (less than 0.2), indicating a good fit of the model. The confidence ellipses of CON, FOS-1, FOS-2, and FOS-3 overlapped with each other, reflecting that the bacterial community structures of these groups were similar.

These findings collectively indicate that dietary supplementation with FOS has limited impact on the alpha-diversity, beta-diversity, and microbial community structure of cecal microbiota in healthy growing rabbits, suggesting that FOS does not disrupt the established intestinal bacterial equilibrium in rabbits.

As shown in Figure 7, the cecal flora of each group is primarily dominated by *Firmicutes*, *Bacteroidetes*, and *Verrucomicrobia* at the phylum level; *Clostridia*, *Bacteroidia*, and *Bacilli* at the class level; *Oscillospirales*, *Bacteroidales*, and *Bacilli* at the order level; and *Lachnospiraceae*, *Ruminococcaceae*, and *Oscillospiraceae* at the family level. Across these three levels, the highest relative abundance was observed in the first-mentioned category for each group except the FOS-3 group at the order level. *Bacteroidales* has the highest relative abundance in the FOS-3 groups at the order level. The histological analyses are summarized in Table 6. No significant differences were observed among these groups. These results revealed that the addition of FOS does not significantly alter the composition of the gut microbiota but rather helps maintain a healthy and balanced microbial community.

As shown in Figure 8, *Lachnospiraceae*_NK4A136_group (potentially beneficial bacteria associated with intestinal barrier function) were enriched in the FOS-1 and FOS-2 groups. *Monoglobus*, which is related to pectin degradation, was enriched in the FOS-2 group. *Ruminococcus* and *Mucinophilic Akkermansia* were enriched in the FOS-3 group. The results highlight distinct abundance patterns of gut microbiota related to intestinal barrier function, pectin degradation, and digestion across groups with different concentrations of FOS treatment. The enhanced populations of specific bacteria may improve gut health and metabolic processes.

## 3. Discussion

Oligofructose (FOS) is a commonly used prebiotic which can promote the growth and metabolism of probiotics, thereby benefiting the health of the host [19,20,21]. However, different doses of oligofructose exhibit varying effects on farmed animals. Excessive concentrations of FOS supplementation can adversely affect animal health [18]. Therefore, the dosage of prebiotics is critical, and excessive doses and underdoses can cause economic losses. This study aims to explore the effects of different FOS supplementation doses on the growth and health of rabbits and provide a basis for determining their optimal feeding level.

Improving the growth performance of farming animals has important economic value in animal husbandry. In this study, FOS supplementation (particularly 0.6% in the FOS-2 group) significantly increased final body weight and average daily gain and reduced the feed/weight ratio compared to controls (Table 1), highlighting its potential to improve growth efficiency and lower feeding costs. In our study, we also found that FOS (0.6% and 0.9%) could enhance slaughter performance such as cold carcass weight. Additionally, FOS supplementation improved meat quality through higher pH_45min_ and reduced drip loss and lower shear force, indicating better tenderness and flavor. Our findings indicate that dietary supplementation of FOS can enhance growth performance, the feed conversion ratio, and meat quality, which would provide favorable economic benefits in rabbit farming.

In this study, we found that the supplemental of FOS could significantly affect lipid metabolism indicators in plasma. In particular, TC (FOS-1, FOS-2, and FOS-3 groups), TG (FOS-2 and FOS-3 groups), FFA (FOS-2 and FOS-3 groups), LDL-C (FOS-3 group), and HDL-C (FOS-2 and FOS-3 groups) levels were notably lower than those in the CON group (Table 3). Similar lipid-lowering effects of FOS supplemental in humans have been discovered [22,23]. These results may indicate that FOS could regulate lipid metabolism, which provides new insights for developing nutritional strategies to control obesity.

Antioxidant enzymes such as superoxide dismutase (SOD) serve as the primary antioxidant defense, which could reduce oxidative stress. Additionally, the content of malondialdehyde (MDA) is often used as a biomarker to measure the level of antioxidant damage in animals as MDA is the end product of lipid peroxidation reactions [24]. Our results showed that FOS enhanced antioxidant capacity via elevated T-AOC and SOD levels, alongside reduced MDA (Table 2), which demonstrated that FOS could improve resilience to stressors of animals under modern intensive farming conditions. These results, which are consistent with studies in rats and dairy cows [25,26], suggested FOS might mitigate oxidative damage by curbing lipid peroxidation and scavenging free radicals. We also found that the inclusion of oligofructose in the diet resulted in higher blood levels of calcium, magnesium, and iron (Table 2). This finding indicated that FOS could enhance mineral absorption. However, the concentrations of acetic acid in cecum were notably lower in the FOS-3 group compared to the CON group (Figure 4), hinting at a potential inhibitory effect of excessive oligofructose on cecal fermentation. While FOS improves resilience to stressors (e.g., weaning, disease), excessive supplementation may paradoxically impair antioxidant function [18]. Thus, optimizing FOS dosage is critical to balancing its benefits in lipid regulation, antioxidant enhancement, and stress resistance.

Immunoglobulins, including IgA, IgG and IgM, serve as critical indicators of humoral immunity to resist pathogens and prevent intestinal diseases [27]. As shown in Figure 1A, the FOS supplemental in the diet could significantly increase the serum levels of IgA, IgG, and IgM in growing rabbits compared to the control group. Among these groups, the 0.6% supplementation group exhibited the optimal enhancement of humoral immunity. Interleukin-2 (IL-2) and interleukin-1α (IL-1α) are involved in numerous normal physiologic processes and contribute to the modulation of various inflammatory diseases [28]. The upregulation of these factors in the intestine indicated an activated immune system, potentially accelerating phagocytosis and pathogen clearance, ultimately enhancing the immune defense capabilities of the organism. Secretory immunoglobulin A (sIgA) is produced by intestinal lymphoid tissue plasma cells and is crucial for mucosal immunity. Its upregulation indicated an enhancement of the intestinal immune system, which contributed to gut health and may prevent the occurrence of intestinal diseases. The 0.3% and 0.6% FOS supplementation groups exhibited upregulated critical immune factors (Figure 1). Our findings suggested that suitable FOS would activate systemic and mucosal immune responses that promote pathogen clearance and gut health.

Immune organ indices are often employed as indicators of immune status, with an increase suggesting enhanced immunity [29]. Our results demonstrate that dietary supplementation with 0.3% FOS (FOS-1 group) significantly enhances the liver index and kidney index in rabbits, while having no notable effect on other immune organ indices (Figure 2). However, raising the FOS level to 0.6% or 0.9% weakens its positive effects on the immune system organs. Combining our results presented above, we reasonably speculate that the immune regulatory effects of FOS on healthy rabbits are primarily manifested in maintaining intestinal health and improving animal feed conversion efficiency by enhancing gut immunity and humoral immunity, as well as activating the immune system.

Given that the primary target of FOS may lie in the intestine, we further evaluated its impact on the tissue structure and functionality of the small intestine. Increases in villi height, length, and V/C value, along with a rise in goblet cell number (secreting mucus for epithelial protection and repair [30,31]), were observed in the duodenum, jejunum, and ileum of the experimental groups compared with the control group, except for villus thickness (Table 3). The morphological integrity of the small intestine is vital for nutrient absorption and transport. Our results indicate that prebiotics like FOS could optimize small intestine microstructure and facilitate efficient nutrient digestion and absorption. We also found that dietary supplementation with FOS exhibited higher activities of cellulase, lipase, and total protease compared to the CON group (Table 4). Among these groups, the 0.6% supplementation level achieved the most pronounced enhancement in digestive enzyme activity. Our findings indicated that FOS supplementation enhances intestinal morphology and digestive enzyme activities, promoting more efficient nutrient digestion and absorption processes, thereby enhancing the growth performance of growing rabbits.

A previous study proved that prebiotics could promote animal growth by selectively enhancing beneficial bacterial colonization [32]. To further investigate the impact of FOS on the gut microbiota, we conducted 16S rRNA sequencing analysis on cecal contents. Table 5 and Figure 6 indicate that the supplementation of FOS does not disrupt the gut microbial flora balance in healthy rabbits. However, supplementation with FOS can alter the abundance of specific gut microbiota to maintain a healthy microbial community. Our findings revealed that FOS could enrich *Akkermansia mucinophilic* (bacteria associated with intestinal barrier function), *Lachnospiraceae*_NK4A136_group (potentially beneficial bacteria), *Monoglobus* (pectin-degrading bacteria), and *Ruminococcus* (Figure 8), suggesting its potential roles in promoting intestinal mucosal health, enhancing feed digestibility, and facilitating the growth of animals. These discoveries indicate that FOS can promote and maintain intestinal microbial homeostasis, which is crucial for the physiological development of rabbits [33,34]. However, since we used healthy rabbits in this study, the effects of FOS on treating intestinal diseases and facilitating the restoration of disrupted gut microbiota balance were not demonstrated. Future studies will employ disease-challenge models (such as pathogen or stress-induced enteritis) to mechanistically evaluate FOS efficacy in maintaining intestinal health and microbiota balance. Approaches such as fecal microbiota transplantation (FMT) and targeted antibiotic depletion would also be valuable for further elucidating the proposed mechanisms.

In this study, we employed growing rabbits as an experimental model to systematically investigate the effects of FOS on growing rabbits (Figure 9). The results indicated that the dietary FOS supplement would enhance the antioxidant capacity of rabbits, effectively counteracting stress responses induced by adverse external environments, thereby improving meat quality and slaughter performance. Furthermore, FOS contributes to the overall health status and feed conversion efficiency of rabbits by influencing humoral and intestinal immunity, maintaining intestinal morphology and modulating the gut microbiota among other aspects. Consistent with previous research findings [18], we believe that while FOS offers multiple benefits such as antioxidant activity, stress resistance, and intestinal health regulation, excessive use may diminish these effects or even exert adverse impacts. In this study, the highest dose of FOS (0.9%) showed potential adverse effects on cecal fermentation and organ indices, but no negative effects on clinical signs, poor growth, or organ pathology were observed. Higher FOS supplementation would also increase costs. Therefore, we recommend that the addition level in meat rabbits should not exceed 0.9%. Based on the outcomes of this experiment, we determined that an appropriate concentration of FOS in the feed for growing rabbits is 0.6%. In future studies, we aim to further explore the specific roles of FOS in animals subjected to intestinal disease challenges, with the goal of uncovering deeper insights into its mechanisms of antioxidant activity, stress resistance, and intestinal health regulation.

## 4. Materials and Methods

### 4.1. Animal Ethics

The study was approved by the Fujian Agriculture and Forestry University Animal Care and Use Committee (Approval code: PZCASFAFU23121) on 15 January 2023.

### 4.2. Animals and Housing Conditions

The trial was carried out at the Rabbit Farm of Fujian Agriculture and Forestry University (Fuzhou, China) during July–September. FOS (purity > 95%, HPLC) was provided by National Engineering Research Center of Juncao Technology (Fuzhou, China). A total of 100 female Dehua black rabbits (34 d of age) were selected from Dehua Jisheng Black Rabbit Breeding Co. Ltd. (Quanzhou, China). All rabbits were allocated into 20 modules (90 cm long × 54 cm wide × 120 cm high) with 5 rabbits per module until slaughtering (106 days of age). Each module was equipped with a plastic slat floor (hole dimensions: 70 mm long × 10 mm wide; distance between holes: 7 mm), a manual feed distribution trough, and two automatic nipple drinkers.

### 4.3. Preparation of Experimental Diets

Four iso-nitrogenous (15.5% crude protein), equal fiber content (12% crude fiber), iso-lipidic (3.5% ether extract), and iso-caloric (260 kcal digestible energy/100 g) practical diets, namely CON (control with 0% oligofructose), FOS-1 (0.3% oligofructose), FOS-2 (0.6% oligofructose), and FOS-3 (0.9% oligofructose), were formulated and prepared (Table 7). According to that formulation, the pre-weighed ingredients were thoroughly mixed with water to form a dough, which was then autoclaved at 121 °C for 20 min. After cooling, heat-sensitive additives were mixed uniformly. A pellet mill was used to extrude the mixture into uniform pellets with a diameter of 2.0 mm. The pellet mill extrudes the material through the die holes of a ring die, and the formed strands are cut into pellets by cutting blades. The pellets are then dried at 45 °C until the moisture content is below 10%. The dried feed pellets are hermetically sealed and stored under refrigerated conditions at 4 °C.

### 4.4. Experimental Design

The rabbits were fed four experimental diets (5 replicates per diet for the trial, 5 rabbits per replicate) based on a completely randomized experimental design with four dietary inclusion levels of FOS (CON: 0%; FOS-1: 0.3%; FOS-2: 0.6%; FOS-3: 0.9%). No antibiotics or coccidiostats were included in the diets or administered via water during the trial.

The experimental diets were fed from weaning (34 days of age) to slaughter (106 days of age). The pre-feeding period lasted for seven days. Immediately after the pretrial period, fecal samples from each experimental group were collected. The fecal samples were used to determine fecal energy (FE) via an oxygen bomb calorimeter (RF-C7000, Ruifang Ltd., Changsha, China) for calculating the digestible energy (DE) of the feed. Throughout the trial, the animals had free access to feed and fresh water. Individual live weights were recorded twice a week to closely monitor rabbit growth and promptly detect any abnormal weight changes and health problems, while pen feed intake was measured daily. The health status of the rabbits was monitored daily. During the trial, no rabbits were excluded due to health problems.

### 4.5. Growth Performance and Slaughtering

Feed intake was recorded daily. At 106 days of age, after a 4 h fasting period, all rabbits in the trial were weighed at the experimental farm before loading to determine the final live weight. Then, 40 rabbits (2 rabbits per replicate, 10 rabbits per diet), representative of the corresponding experimental groups in terms of average rabbit live weight and SD, were individually weighed, stunned using electro anesthesia, and slaughtered by jugulation. After 24 h of chilling, the carcasses were weighed to calculate the individual dressing percentage.

### 4.6. Sample Collection

After slaughtering, blood was immediately taken from the heart to obtain samples with a more consistent mixture of systemic blood. Once collected, the blood was placed into lithium heparin tubes. The plasma was directly obtained by centrifugation (Thermo Scientific SL, 16R, Thermo Fisher Scientific, Waltham, MS, USA) at 4000× *g* for 10 min, at 4 °C, and immediately stored at −20 °C. The thymus, spleen, liver, and kidney were weighed to calculate the immune organ development index (g/kg).

The dorsal and leg muscle was selected as a sample for analysis. The pH of the muscle was measured by a hand-held pH meter (PHSJ-3F, Shanghai, China). Drip loss was determined by hanging a section from a 5 cm × 3 cm × 2 cm muscle sample in a sealed plastic bag for 24 h at 4 °C. Shear force was determined by cutting a 3 cm × 1 cm × 1 cm piece of muscle and testing it with a digital muscle tenderness tester (C-LM3B, Tenovo, Beijing, China).

The jejunum and ileum were excised and cleaned with saline. The mucosa was scraped off, and these samples were stored at −80 °C. Inflammatory factors (including IL-1α, IL-2, and sIgA) were determined using ELISA kits (Shanghai Liquid Quality Testing Technology Co., Ltd., Shanghai, China) according to the instructions of the manufacturer. Duodenal, jejunal, and ileal segments were fixed in 4% paraformaldehyde for histological examination. Then the villus height (VH), crypt depth (CD), VH/CD ratio, and goblet cell (GC) were measured.

Parts of the cecum were finely homogenized in saline using a homogenizer (FSH-2A, Japid Co., Ltd., Qingdao, China) and centrifuged at 8000 rpm for 5 min, at 4 °C, to obtain the supernatants. The activities of cellulase, lipase, and total protease were measured by commercial assay kits (Shanghai Liquid Quality Testing Technology Co., Ltd., Shanghai, China) according to the instructions of the manufacturer. Samples of cecal contents were aseptically collected in 2 mL Eppendorf tubes and stored at −80 °C until analysis.

### 4.7. Chemical and Biochemical Analysis

The plasmic biochemical parameters were determined using a fully automatic biochemical analysis system (BS-200, Mindray Biomedical Electronics Co., Ltd., Shenzhen, China). A comprehensive set of biochemical parameters were measured, including total protein (TP), triglyceride (TG), high-density lipoprotein cholesterol (HDL-C), total cholesterol (TC), and low-density lipoprotein cholesterol (LDL-C).

Free fatty acid (FFA) levels were tested by commercial test kit (Nanjing Jiancheng Technology Co., Ltd., Nanjing, China) according to the instructions of the manufacturer. Total antioxidant capacity (T-AOC), plasmic superoxide dismutase (SOD), and plasmic malondialdehyde (MDA) content were measured by using the commercial assay kits (Shanghai Liquid Quality Testing Technology Co., Ltd., Shanghai, China) according to the instructions of the manufacturer.

IgA, IgG, IgM, IL-1α, IL-2, and sIgA were measured by ELISA kits (Shanghai Liquid Quality Testing Technology Co., Ltd., Shanghai, China) according to the instructions of the manufacturer.

### 4.8. Cecal Fermentation

The cecal contents were collected and diluted with an appropriate volume of PBS buffer in a 1:4 ratio. This mixture was vigorously shaken and blended for 2 min, followed by centrifugation to eliminate solid impurities, yielding a clarified supernatant. The supernatant was then analyzed for short-chain fatty acids, including acetic acid, propionic acid, butyric acid, valeric acid, caproic acid, isobutyric acid, and isovaleric acid, using a gas chromatograph (GC 2010-FID, Shimadzu, Kyoto, Japan). The gas chromatograph system featured a KB-FFAp column from Kromat and a flame ionization detector set to 250 °C. The column was maintained at 80 °C with a nitrogen mobile phase flowing at a rate of 0.81 mL/min.

### 4.9. Cecal Microbiota Analysis

Total DNA of the cecal contents was extracted using the cetyltrimethylammonium bromide (CTAB) lysis buffer method. Then the prepared DNA samples were submitted to Biomarker Technologies Co., LTD. (Beijing, China) for 16S rRNA-based microbiota analysis. The hypervariable V3-V4 region of the 16S rDNA gene was amplified by polymerase chain reaction (PCR). Purified amplicons were pooled in equimolar and paired-end sequences (PE250) on an Illumina platform according to the standard protocols.

For 16S gene sequencing, the raw sequencing reads were processed and quality-checked using the microbial genomics workflow of the CLC Genomics Workbench (version 22.0.2) from the Computational Life Science Center of Qiagen (Hilden, Germany). Samples were filtered based on the number of reads to ensure comparable sequencing coverage. The minimum number of reads was set at 100, and the minimum percentage relative to the median was set at 50%. High-quality reads were clustered into operational taxonomic units (OTUs) at a 97% similarity level. Community composition analysis, indicator species analysis (observed OTUs, rank abundance curve, and dilution curve analysis) and alpha-diversity analysis (Coverage, Ace, Chao1, Shannon index) were performed using QIIME2.

### 4.10. Statistics

A completely randomized experimental design was employed in this study. All data were analyzed using IBM SPSS Statistics software (version 25.0). Data were first subjected to normality testing using the Shapiro–Wilk test. For datasets conforming to normal distribution, Bartlett’s test was employed to assess homogeneity of variance. When homogeneity of variance was confirmed, one-way analysis of variance (ANOVA) was performed; otherwise (heterogeneous variance or non-normal distribution), the non-parametric Kruskal–Wallis test was applied. Unless otherwise stated, all data met the assumption of homogeneity of variance. The CON group (without FOS supplement in diet) was set as the control. When the ANOVA indicated significant differences among groups, Tukey’s multiple range tests were further employed to determine which specific pairs of groups were significantly different. The results are presented as the mean ± standard error (SEM). The 16S rDNA sequencing data were analyzed on the free online platform of Omicsmart tools (https://www.omicshare.com/tools, accessed on 5 December 2023), and the Kruskal–Wallis test was employed for analysis of significant differences. A *p*-value less than 0.05 indicated a statistically significant difference, while a *p*-value less than 0.01 indicated a highly statistically significant difference.

## 5. Conclusions

This study investigated the effects of different oligofructose levels in diets on growing rabbits. The results indicated that oligofructose enhanced growth, slaughter performance, and muscle quality, with optimal benefits observed at 0.6% for growth and lipid metabolism. FOS could also enhance immunity and intestinal morphology. Additionally, FOS improved antioxidant capacity and mineral absorption and maintained intestinal flora balance. Overall, 0.6% oligofructose was identified as the optimal dietary addition for growing rabbits. The findings of this study provide valuable insights for developing FOS as an efficient growth-promoting additive and provide a reliable basis for its optimum additive amount in rabbit farming.

## Figures and Tables

**Figure 1 ijms-26-08694-f001:**
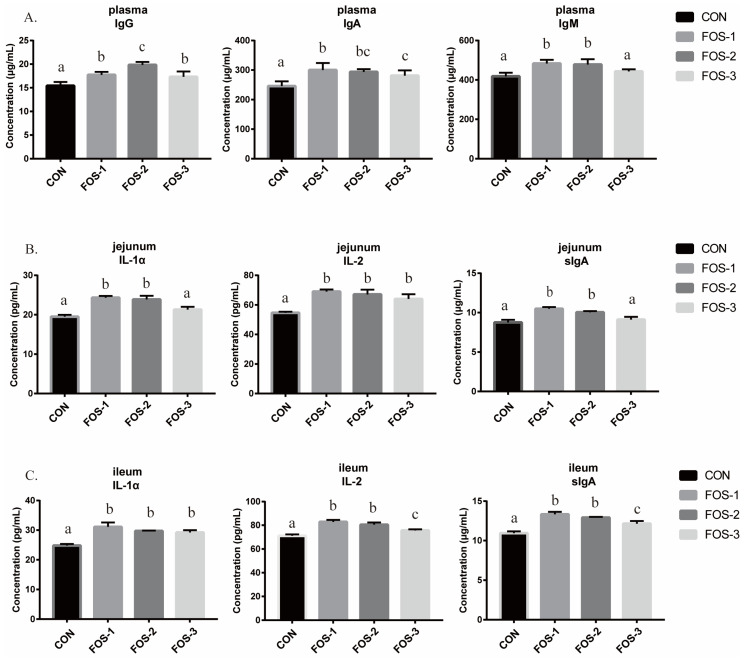
The immune factor levels ((**A**) in the plasma, (**B**) in the jejunum, (**C**) in the ileum). The error bar indicates SEM (data are from Table A1). Different letters indicate significant differences (*p* ≤ 0.05). IgG: Immunoglobulin G; IgA: Immunoglobulin A; IgM: Immunoglobulin M; IL-1α: Interleukin-1α; IL-2: Interleukin-2; sIgA: Secretory immunoglobulin A. n = 3.

**Figure 2 ijms-26-08694-f002:**
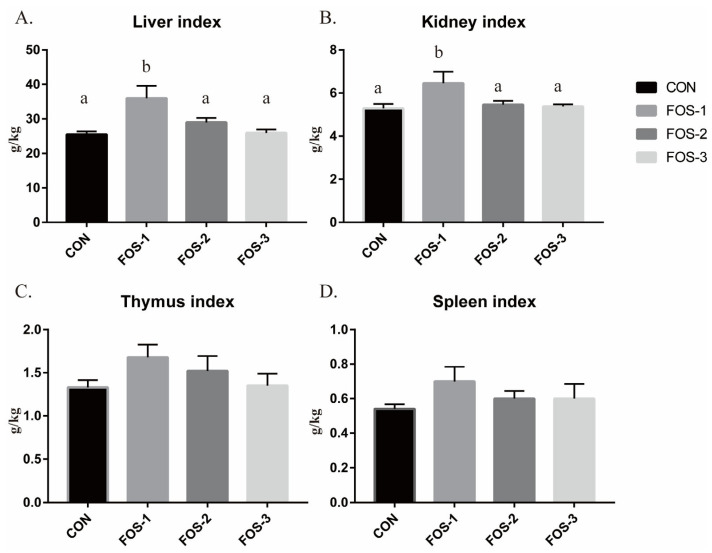
The immune organs index ((**A**) liver index, (**B**) kidney index, (**C**) thymus index, (**D**) spleen index). The error bar indicates standard SEM (data are from Table A2). Different letters indicate significant differences (*p* ≤ 0.05). n = 5.

**Figure 3 ijms-26-08694-f003:**
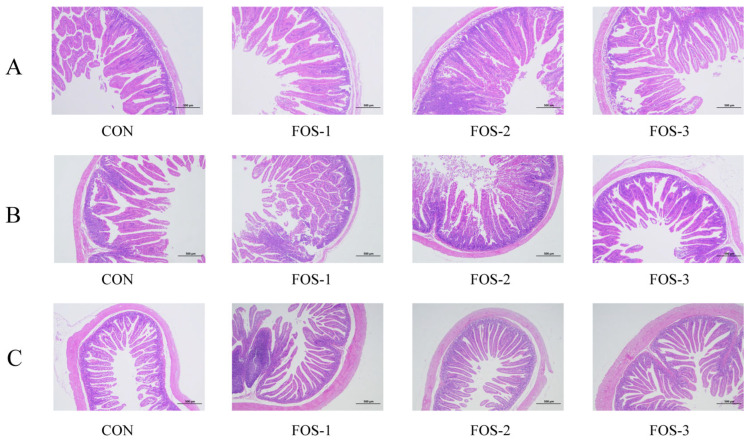
Microscopic images of the HE slide (100×) of duodenum (**A**), jejunum (**B**), and ileum (**C**) from the CON group, FOS-1 group, FOS-2 group, and FOS-3 group. The histological analyses of muscle are shown in Table 3.

**Figure 4 ijms-26-08694-f004:**
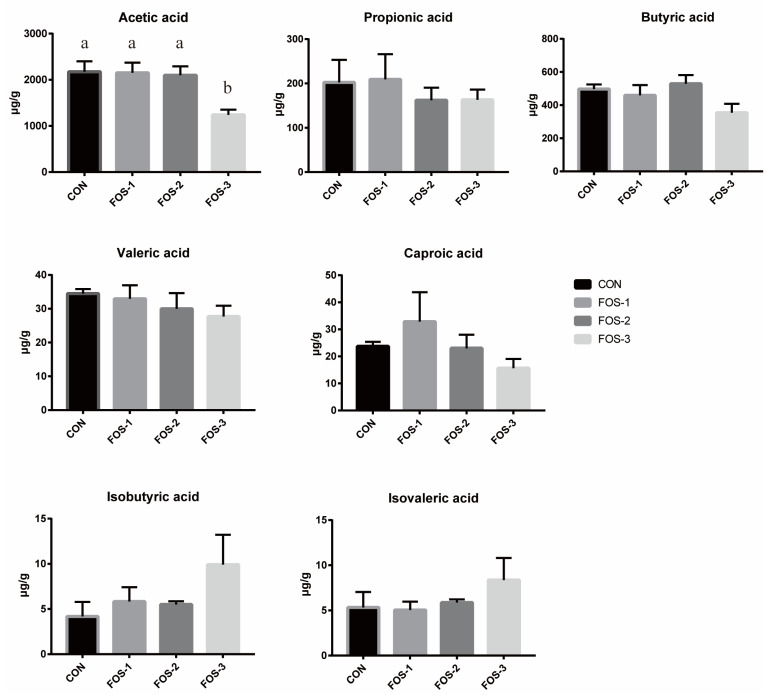
Short-chain fatty acid profile in cecal contents. The error bar indicates standard SEM (data are from Table A3). Different letters indicate significant differences (*p* ≤ 0.05). n = 5.

**Figure 5 ijms-26-08694-f005:**
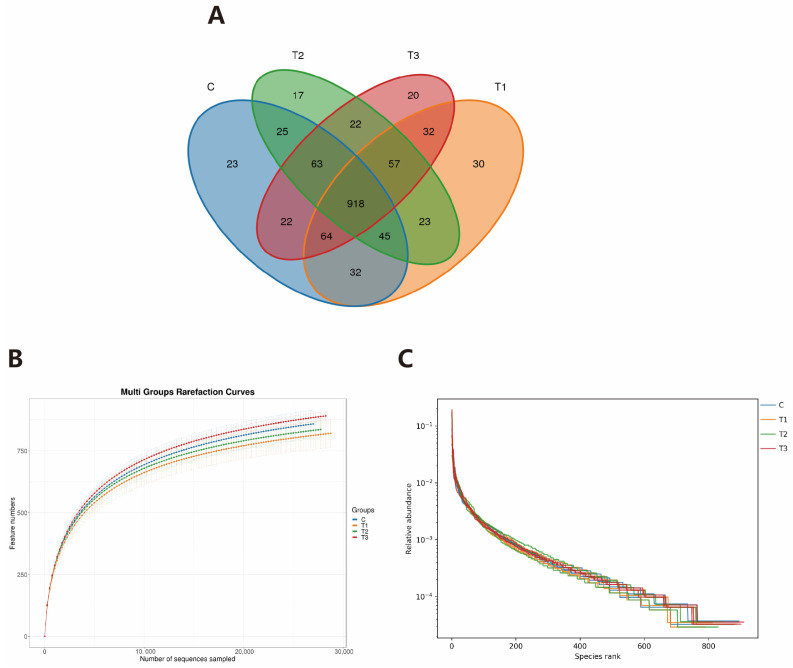
(**A**) The OTU Venn diagram of microflora in the cecum. (**B**) Dilution curve of cecum microorganism sequencing. (**C**) The OTU abundance levels of microflora in the cecum. C: CON group; T1: FOS-1 group; T2: FOS-2 group; T3: FOS-3 group. n = 3.

**Figure 6 ijms-26-08694-f006:**
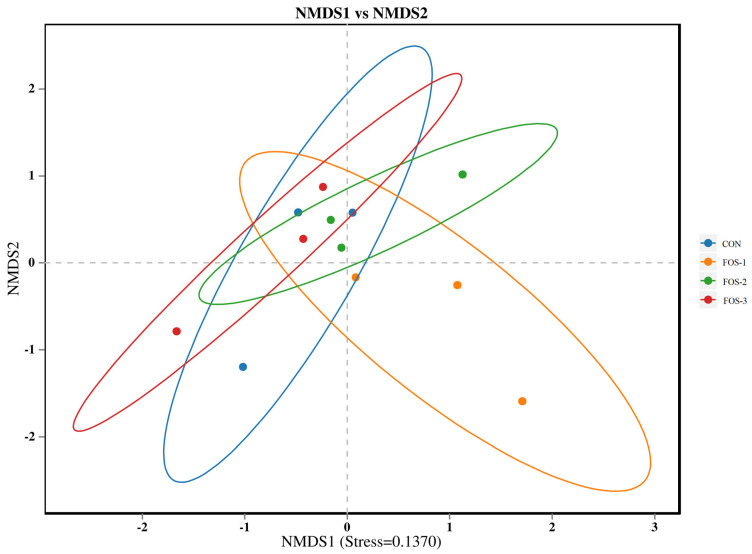
NMDS analysis diagram. The stress value is a measure of distortion. It is generally considered that when stress is less than 0.2, the results of NMDS analysis are reasonably reliable. n = 3.

**Figure 7 ijms-26-08694-f007:**
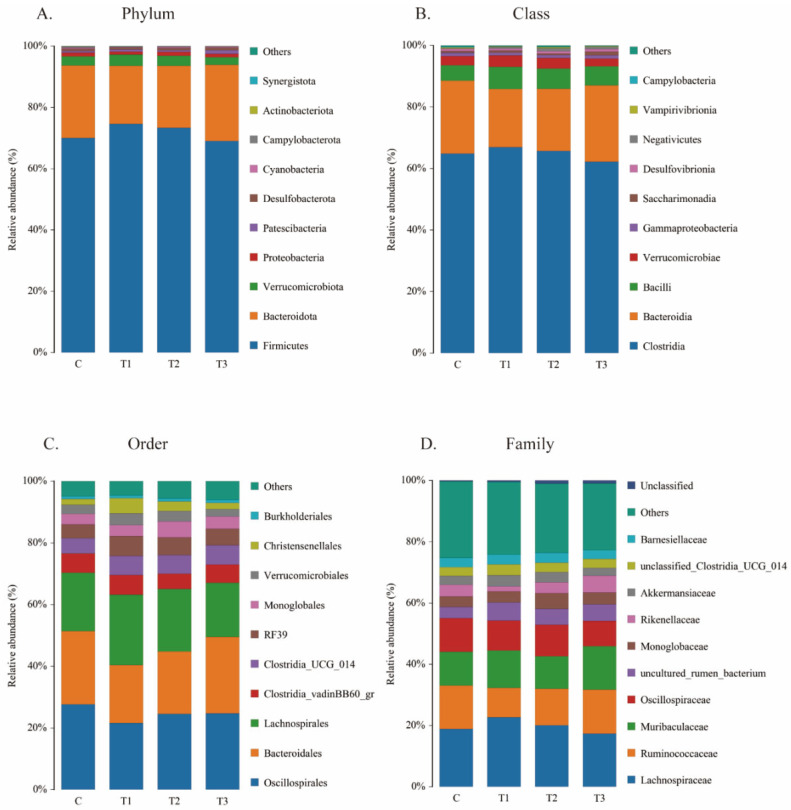
Relative abundance of operational taxonomic units (OTUs) in the cecal microbiota of rabbits at (**A**) phylum level, (**B**) class level, (**C**), order level, and (**D**) family level. C: CON group; T1: FOS-1 group; T2: FOS-2 group; T3: FOS-3 group. n = 3.

**Figure 8 ijms-26-08694-f008:**
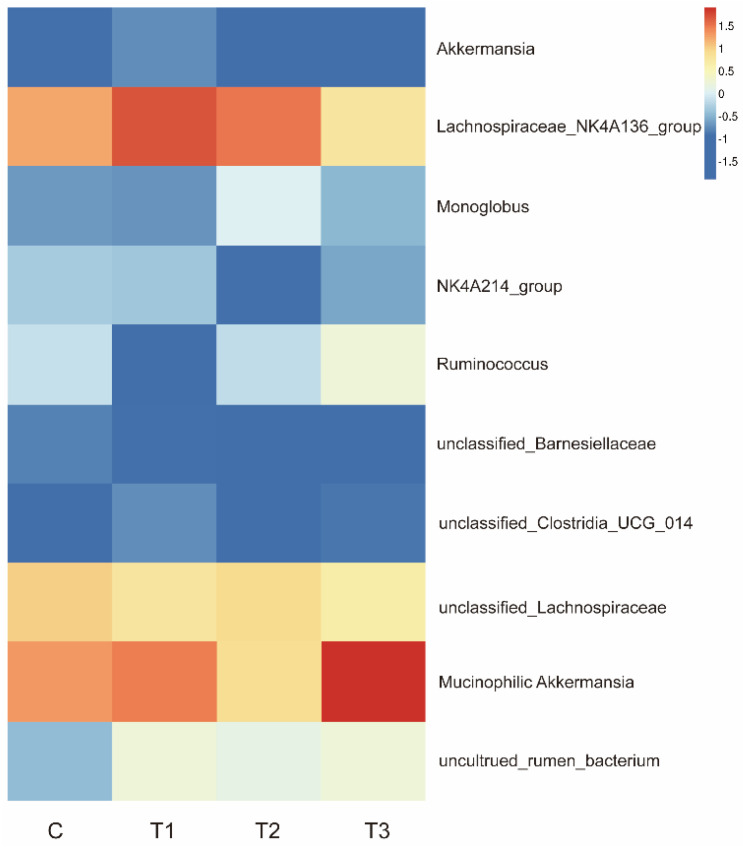
Heatmap depicting functional annotation clusters of cecal microbiota. Species with high and low abundances are aggregated into clusters, with color gradients and degrees of similarity employed to reflect the similarities and differences in community composition across multiple samples. Redder color indicates higher degree of aggregation. C: CON group; T1: FOS-1 group; T2: FOS-2 group; T3: FOS-3 group. n = 3.

**Figure 9 ijms-26-08694-f009:**
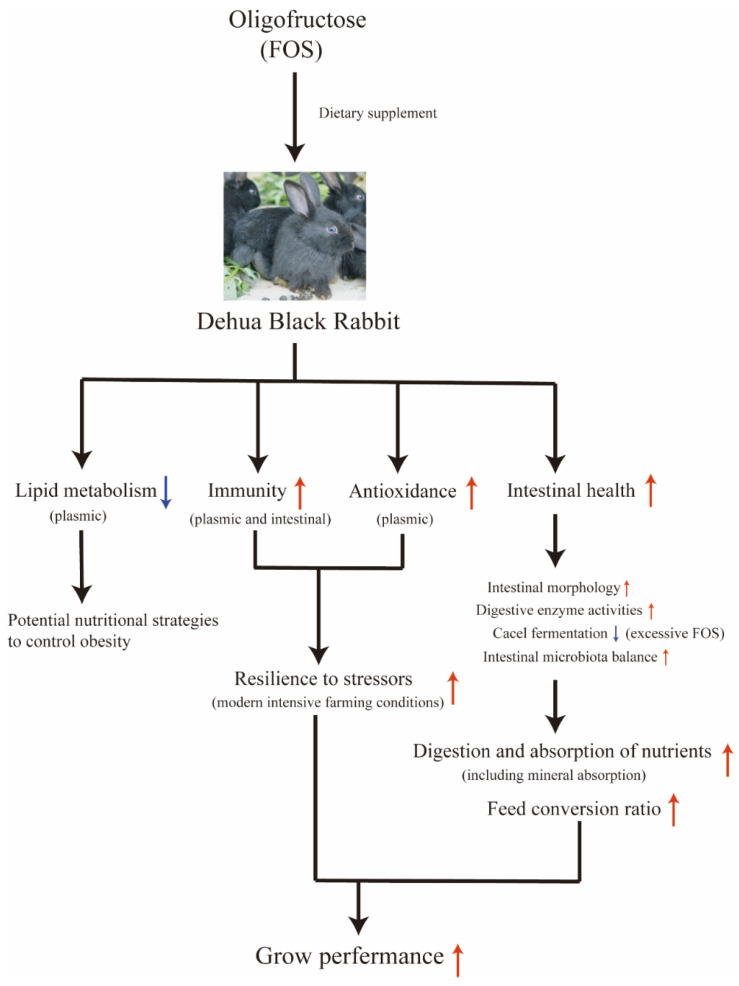
Effects of dietary FOS supplement on growing rabbits (Dehua black rabbits). Red arrows indicate upregulation or enhancement. Blue arrows indicate downregulation or inhibition.

**Table 1 ijms-26-08694-t001:** Growth performance, carcass trait, and meat quality.

Items	Groups				*p*-Value
	CON	FOS-1	FOS-2	FOS-3	
IBW (kg)	1.13 ± 0.09	1.05 ± 0.09	1.15 ± 0.04	1.18 ± 0.08	0.52
FBW (kg)	2.85 ± 0.07 ^a^	2.87 ± 0.09 ^a^	3.04 ± 0.05 ^b^	2.99 ± 0.08 ^ab^	0.01
ADFI (g)	136.88 ± 1.39	133.29 ± 2.11	138.45 ± 0.59	134.19 ± 1.14	0.85
ADG (g)	23.85 ± 0.69 ^a^	25.28 ± 0.78 ^b^	26.39 ± 0.65 ^b^	25.21 ± 0.44 ^b^	<0.01
F/G	5.76 ± 0.17 ^a^	5.29 ± 0.16 ^b^	5.26 ± 0.13 ^b^	5.33 ± 0.09 ^b^	0.05
Cold carcass weight (kg)	1.35 ± 0.07 ^a^	1.38 ± 0.04 ^a^	1.57 ± 0.04 ^b^	1.55 ± 0.04 ^b^	<0.01
Dressing percentage (%)	49.34 ± 0.94	48.73 ± 0.55	49.77 ± 1.21	49.68 ± 0.87	0.85
Dorsal muscle					
pH_45min_	6.42 ± 0.04 ^a^	6.52 ± 0.08 ^ab^	6.67 ± 0.04 ^b^	6.54 ± 0.03 ^ab^	0.02
pH_24h_	5.99 ± 0.04	5.97 ± 0.01	5.92 ± 0.04	5.93 ± 0.00	0.26
Drip loss rate (%)	31.24 ± 1.13 ^a^	24.20 ± 1.21 ^b^	22.31 ± 0.65 ^b^	11.93 ± 0.19 ^c^	<0.01
Shear force (N)	32.31 ± 0.59 ^a^	30.34 ± 0.55 ^b^	28.56 ± 0.52 ^b^	23.46 ± 0.82 ^c^	<0.01
Leg muscle					
pH_45min_	6.50 ± 0.05	6.51 ± 0.11	6.75 ± 0.07	6.62 ± 0.05	0.09
pH_24h_	6.11 ± 0.07	6.06 ± 0.02	6.05 ± 0.06	6.02 ± 0.04	0.66
Drip loss rate (%)	30.45 ± 0.23 ^a^	16.27 ± 0.75 ^b^	15.42 ± 0.41 ^b^	6.83 ± 0.19 ^c^	<0.01
Shear force (N)	24.98 ± 0.42 ^a^	22.12 ± 0.38 ^b^	20.85 ± 0.24 ^b^	18.15 ± 0.60 ^c^	<0.01

IBW, initial body weight; FBW, final body weight; ADFI, average daily feed intake; ADG, average daily gain; F/G, feed/gain ratio. The results are presented as the mean ± SEM. Different letters indicate significant differences (*p* ≤ 0.05). For IBM, FBM, ADFI, ADG, and F/G, n = 25; for others, n = 5.

**Table 2 ijms-26-08694-t002:** Plasmic biochemical parameter, antioxidant capacity, and mineral contents.

Items	Groups				*p*-Value
	CON	FOS-1	FOS-2	FOS-3	
TC (mmol/L)	2.46 ± 0.03 ^a^	2.31 ± 0.04 ^b^	2.19 ± 0.02 ^b^	1.73 ± 0.05 ^c^	<0.01
TG (mmol/L)	1.13 ± 0.02 ^a^	1.08 ± 0.02 ^ab^	0.98 ± 0.03 ^b^	0.80 ± 0.02 ^c^	<0.01
HDL-C (mmol/L)	1.73 ± 0.06 ^a^	1.60 ± 0.07 ^a^	1.18 ± 0.03 ^b^	1.01 ± 0.01 ^c^	<0.01
LDL-C (mmol/L)	0.92 ± 0.04 ^a^	0.90 ± 0.06 ^a^	0.88 ± 0.03 ^a^	0.73 ± 0.02 ^b^	0.03
FFA (mmol/L)	411.01 ± 8.57 ^a^	400.82 ± 7.93 ^a^	356.81 ± 5.91 ^b^	318.67 ± 5.87 ^c^	<0.01
T-AOC (U/mL)	19.35 ± 0.45 ^a^	22.36 ± 0.25 ^b^	23.45 ± 0.08 ^c^	25.55 ± 0.36 ^d^	<0.01
MDA (nmol/mL)	5.81 ± 0.07 ^a^	5.04 ± 0.06 ^b^	4.01 ± 0.05 ^c^	4.53 ± 0.03 ^d^	<0.01
SOD (U/mL)	16.71 ± 0.66 ^a^	21.32 ± 0.32 ^b^	29.94 ± 0.49 ^c^	20.94 ± 0.18 ^b^	<0.01
Ca (mmol/L)	2.25 ± 0.09 ^a^	2.40 ± 0.18 ^a^	2.80 ± 0.10 ^b^	2.48 ± 0.06 ^ab^	0.05
Mg (mmol/L)	0.45 ± 0.03 ^a^	0.68 ± 0.02 ^b^	0.89 ± 0.03 ^c^	1.00 ± 0.01 ^d^	<0.01
Fe (mmol/L)	19.34 ± 0.34 ^a^	22.81 ± 0.97 ^b^	25.15 ± 0.89 ^b^	24.46 ± 1.14 ^b^	<0.01

TC: total cholesterol; TG: triglyceride; HDL-C: high-density lipoprotein cholesterol; LDL-C: low-density lipoprotein cholesterol; FFA: Free fatty acid. T-AOC: total antioxidant capacity; MDA: malondialdehyde; SOD: superoxide dismutase. The results are presented as the mean ± SEM. Different letters indicate significant differences (*p* ≤ 0.05). n = 3.

**Table 3 ijms-26-08694-t003:** Intestinal morphologic structure.

Items	Groups				*p*-Value
	CON	FOS-1	FOS-2	FOS-3	
duodenum					
Villus height (μm)	0.91 ± 0.21	1.14 ± 0.03	1.22 ± 0.08	1.18 ± 0.10	0.08
Crypt depth (μm)	0.11 ± 0.01	0.11 ± 0.02	0.11 ± 0.02	0.11 ± 0.02	0.94
V/C	8.16 ± 2.28 ^a^	11.39 ± 2.48 ^ab^	12.00 ± 3.69 ^b^	10.73 ± 3.79 ^ab^	0.05
Villus length (μm)	1.17 ± 0.09 ^ab^	1.05 ± 0.18 ^a^	0.91 ± 0.08 ^a^	1.41 ± 0.26 ^b^	0.04
Villus thickness (μm)	0.04 ± 0.00	0.05 ± 0.02	0.04 ± 0.00	0.04 ± 0.00	0.44
Goblet cells (cells)	14.73 ± 6.58	16.60 ± 6.22	10.73 ± 2.72	21.00 ± 6.36	0.25
Goblet cell density (cells/mm)	8.02 ± 0.91	16.46 ± 8.19	12.44 ± 2.54	12.13 ± 2.12	0.22
jejunum					
Villus height (μm)	0.73 ± 0.20	0.82 ± 0.18	0.71 ± 0.38	1.15 ± 0.30	0.27
Crypt depth (μm)	0.14 ± 0.00 ^a^	0.15 ± 0.03 ^a^	0.09 ± 0.01 ^b^	0.09 ± 0.01 ^b^	<0.01
V/C	5.20 ± 1.53 ^a^	5.62 ± 0.11 ^a^	7.99 ± 4.92 ^ab^	12.45 ± 3.09 ^b^	0.05
Villus length (μm)	0.80 ± 0.23	0.77 ± 0.46	0.95 ± 0.27	1.12 ± 0.19	0.53
Villus thickness (μm)	0.03 ± 0.01	0.03 ± 0.01	0.04 ± 0.02	0.03 ± 0.01	0.96
Goblet cells (cells)	13.00 ± 1.00	13.60 ± 6.47	14.60 ± 2.03	20.60 ± 5.10	0.19
Goblet cell density (cells/mm)	13.87 ± 1.22 ^a^	18.74 ± 5.05 ^ab^	23.99 ± 2.31 ^b^	20.79 ± 3.82 ^b^	0.04
ileum					
Villus height (μm)	0.37 ± 0.16 ^a^	0.65 ± 0.01 ^b^	0.57 ± 0.05 ^b^	0.70 ± 0.04 ^b^	<0.01
Crypt depth (μm)	0.10 ± 0.01	0.10 ± 0.02	0.09 ± 0.01	0.10 ± 0.01	0.77
V/C	3.71 ± 1.68 ^a^	6.69 ± 1.44 ^b^	6.04 ± 0.80 ^b^	6.77 ± 0.28 ^b^	0.04
Villus length (μm)	0.73 ± 0.33 ^a^	1.03 ± 0.06 ^ab^	0.76 ± 0.05 ^a^	1.21 ± 0.12 ^b^	0.04
Villus thickness (μm)	0.03 ± 0.00	0.03 ± 0.01	0.04 ± 0.01	0.03 ± 0.00	0.10
Goblet cells (cells)	6.13 ± 1.86 ^a^	12.40 ± 1.4 ^b^	13.60 ± 5.64 ^b^	24.67 ± 7.09 ^c^	<0.01
Goblet cell density (cells/mm)	16.08 ± 8.96	13.49 ± 1.38	20.18 ± 4.73	22.91 ± 8.71	0.38

The results are presented as the mean ± SEM. Different letters indicate significant differences (*p* ≤ 0.05). V/C: villus height/crypt depth ratio. n = 3.

**Table 4 ijms-26-08694-t004:** Intestinal enzyme activities.

Items	Groups				*p*-Value
	CON	FOS-1	FOS-2	FOS-3	
Cellulase (U/g)	392.04 ± 1.64 ^a^	429.31 ± 3.69 ^b^	466.36 ± 6.57 ^c^	455.87 ± 3.67 ^d^	<0.01
Lipase (U/g)	42.14 ± 0.32 ^a^	46.41 ± 0.55 ^b^	52.50 ± 1.25 ^c^	45.59 ± 0.65 ^b^	<0.01
Total protease (U/kg)	114.93 ± 2.68 ^a^	157.12 ± 2.56 ^b^	176.28 ± 7.20 ^c^	140.42 ± 2.98 ^d^	<0.01

The results are presented as the mean ± SEM. Different letters indicate significant differences (*p* ≤ 0.05). n = 5.

**Table 5 ijms-26-08694-t005:** Cecal microbial alpha-diversity.

Items	Groups				*p*-Value
	CON	FOS-1	FOS-2	FOS-3	
Coverage	0.99	0.99	0.99	0.99	/
ACE	961.04 ± 27.43	909.95 ± 34.56	927.65 ± 12.46	988.04 ± 9.92	0.16
Chao1	965.98 ± 31.43	931.15 ± 31.34	942.23 ± 12.49	1008.35 ± 4.50	0.15
Shannon	7.39 ± 0.14	7.36 ± 0.17	7.47 ± 0.31	7.40 ± 0.09	0.96

The results are presented as the mean ± SEM, except “coverage”. Data were analyzed using Kruskal–Wallis test. Different letters indicate significant differences (*p* ≤ 0.05). n = 3.

**Table 6 ijms-26-08694-t006:** The relative abundance (%) of OTUs in the cecal microbiota.

Items	Groups				*p*-Value
	CON	FOS-1	FOS-2	FOS-3	
phylum					
*Firmicutes*	69.99 ± 0.11	75.03 ± 3.39	73.83 ± 7.50	68.84 ± 4.00	0.76
*Bacteroidota*	23.66 ± 0.81	18.63 ± 2.47	19.73 ± 7.06	24.99 ± 3.32	0.46
*Verrucomicrobiota*	3.03 ± 0.83	3.56 ± 1.02	3.29 ± 0.72	2.49 ± 0.62	0.90
*Proteobacteria*	1.13 ± 0.25	0.98 ± 0.10	1.26 ± 0.33	1.15 ± 0.25	0.90
class					
*Clostridia*	64.88 ± 0.84	67.27 ± 3.16	66.13 ± 6.63	61.97 ± 4.43	0.76
*Bacteroidia*	23.66 ± 0.81	18.63 ± 2.47	19.73 ± 7.06	24.99 ± 3.32	0.46
*Bacilli*	4.97 ± 0.62	7.25 ± 0.75	6.65 ± 0.91	6.19 ± 0.23	0.15
*Verrucomicrobiae*	3.03 ± 0.83	3.56 ± 1.02	3.29 ± 0.72	2.49 ± 0.62	0.90
order					
*Oscillospirales*	27.62 ± 2.42	21.85 ± 1.67	24.68 ± 0.67	24.63 ± 1.82	0.51
*Bacteroidales*	23.65 ± 0.66	18.50 ± 1.92	19.73 ± 5.76	24.98 ± 2.72	0.46
*Lachnospirales*	19.00 ± 0.51	22.60 ± 1.45	20.41 ± 3.18	17.44 ± 1.21	0.24
*Clostridia*	11.18 ± 0.69	12.65 ± 1.08	11.10 ± 1.37	12.21 ± 0.73	0.79
family					
*Lachnospiraceae*	18.89 ± 0.50	22.53 ± 1.46	20.29 ± 3.18	17.32 ± 1.19	0.24
*Ruminococcaceae*	13.97 ± 2.06	9.71 ± 0.78	11.89 ± 0.28	14.25 ± 1.29	0.28
*Oscillospiraceae*	11.08 ± 1.66	9.90 ± 0.87	10.41 ± 0.74	8.22 ± 0.57	0.49
*Muribaculaceae*	10.64 ± 5.04	11.93 ± 2.59	9.84 ± 5.84	14.41 ± 4.06	0.83

The results are presented as the mean ± SEM. Data were analyzed using Kruskal–Wallis test. Different letters indicate significant differences (*p* ≤ 0.05). n = 3.

**Table 7 ijms-26-08694-t007:** Formulation and proximate composition (on dry matter basis) of the experimental diets.

Items	Experimental Diets ^a^			
	CON	FOS-1	FOS-2	FOS-3
Alfalfa meal	28.5	28.5	28.5	28.5
Corn	21.7	21.7	21.7	21.7
Rapeseed dregs	5.0	5.0	5.0	5.0
Wheat bran	21.4	21.4	21.4	21.4
Soybean meal	8.4	8.4	8.4	8.4
Extruded soybean	5.0	5.0	5.0	5.0
Ca(HCO_3_)_2_	0.7	0.7	0.7	0.7
Methionine	0.1	0.1	0.1	0.1
Lysine	0.2	0.2	0.2	0.2
Wheat meal ^b^	5.0	4.7	4.4	4.1
FOS ^c^	0.0	0.3	0.6	0.9
Premix ^d^	4.0	4.0	4.0	4.0
Nutrient levels ^e^				
Moisture (%)	9.78	10.21	10.04	9.67
Crude Protein (%)	15.48	15.39	15.42	15.34
Crude Fiber (%)	12.25	12.11	12.21	12.10
Ether Extract (%)	3.63	3.55	3.45	3.51
Calcium (%)	0.73	0.65	0.71	0.69
Phosphorus (%)	0.37	0.42	0.35	0.46
Lysine (%)	0.76	0.73	0.76	0.79
Methionine + Cystine (%)	0.45	0.50	0.44	0.47
GE ^f^ (kcal/100 g)	392.10	386.38	384.60	382.11
DE ^g^ (kcal/100 g)	260.60	258.86	259.43	257.12
P:E ^h^ (mg protein/kcal DE)	59.40	59.45	59.44	59.66

Proximate composition is expressed as mean (n = 3). ^a^ CON, diet with 15.5% crude protein, 12% crude fiber, 3.5% ether extract, and 0% FOS; FOS-1, diet with 15.5% crude protein, 12% crude fiber, 3.5% ether extract, and 0.3% FOS; FOS-2, diet with 15.5% crude protein, 12% crude fiber, 3.5% ether extract, and 0.6% FOS; FOS-3, diet with 15.5% crude protein, 12% crude fiber, 3.5% ether extract, and 0.9% FOS. ^b^ Wheat meal serves as a carrier for FOS to ensure uniform mixing. ^c^ FOS (oligofructose, purity > 95%, HPLC) was provided by National Engineering Research Center of Juncao Technology (Fuzhou, China). ^d^ The vitamin and mineral premix supplied Cu 20.0 mg, Fe 75.0 mg, Mn 30.0 mg, Zn 80.0 mg, I 1 mg, Se 0.50 mg, vitamin A 20,000 IU, vitamin D 5000 IU, and vitamin E 50.0 mg per kg of diet. ^e^ Moisture, crude protein (CP), crude fiber (CF), ether extract (EE), calcium (Ca), phosphorus (P), lysine, methionine, and cystine were measured and recorded. CP was determined by the Kjeldahl method; CFs were measured using the filter bag technique; Ca was determined by the ethylenedi-aminetetraacetic acid (EDTA) complexometric titration method; P was measured by spectrophotometry at a wavelength of 400 nm with the molybdenum blue reaction; and amino acids were measured by HPLC. ^f^ GE, gross energy, was measured by an oxygen bomb calorimeter (RF-C7000, Ruifang Co., Ltd., Changsha, China). ^g^ DE, digestible energy = GE − FE. FE (fecal energy) was measured by an oxygen bomb calorimeter (RF-C7000, Ruifang Ltd.,Changsha, China). ^h^ P:E, protein/energy ratio = (CP% × 1000)/DE (kcal/100 g).

## Data Availability

Most of the data generated or analyzed in this study are presented in this published article. Additional data not included here are accessible upon reasonable request to the corresponding author.

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
