# Peer review of "Effects of Oligofructose Supplementation on Growth Performance, Antioxidant Capacity, Immunity, and Intestinal Health in Growing Rabbits"

_ijms, 2025, doi:10.3390/ijms26178694_

Round 1
Reviewer 1 Report
Comments and Suggestions for Authors
The present study aimed at investigating the effects of FOS on growth performance, antioxidant capacity, immunity and intestinal microbial composition in growing rabbits. Overall, the study was well-designed and written in good English. The findings greatly benefit readers in the field of animal nutrition and rabbits husbandary. Only minor revisions were needed to improve the presention of results before accept in IJMS.
- For figure 2, unit should be added for organ index. Also for the result of table 3, unit for goblet cells and goblet cell density. In figure 3, add unit for the result of caproic acid.
- in 2.4, it would be better if the authors could add HE slide of intesine to directly show the difference of intestinal morphology among the four groups.
- for the experiment diets, whether the nutrient contents like CP, CF and EE was the result of calculation or experiment analysis? The authors should stated.
- Since three dose of FOS was used in this study. based on the results, the authors should suggest the best dose in the conclusion.
Author Response
We are grateful for your important suggestions. In accordance with the reviewer concerns, we have revamped whole manuscript. Here are our responses to your comments. Our responses are given directly in a different color (blue). The changes made when revising the manuscript are highlighted in a different color (red).
- For figure 2, unit should be added for organ index. Also for the result of table 3, unit for goblet cells and goblet cell density. In figure 3, add unit for the result of caproic acid.
Response: We thank the reviewer for this valuable suggestion. Units for the organ index (g/kg) have been added to Figure 2 (Line 129). Goblet cells are quantified by counting, hence the unit is “number” and abbreviated as “n”. In Table 3, units for goblet cells (n) and goblet cell density (n/mm) have been added (Line 155). For Figure 4 (originally Figure 3), the unit for caproic acid (μg/g) has been added (Line 174).
- in 2.4, it would be better if the authors could add HE slide of intesine to directly show the difference of intestinal morphology among the four groups.
Response: Thanks for your suggestion. We have added a new figure (Figure 3) showing the HE-stained section of the intestine (Line 152).
- for the experiment diets, whether the nutrient contents like CP, CF and EE was the result of calculation or experiment analysis? The authors should stated.
Response: The nutrient contents (including CP, CF, EE and so on) were the result of experiment analysis. The relevant descriptions have been added in 4.3. Preparation of experimental diets (Line 390-395).
- Since three dose of FOS was used in this study. based on the results, the authors should suggest the best dose in the conclusion.
Response: Based on the outcomes of this experiment, we determined that an appropriate concentration of FOS in the feed for growing rabbits is 0.6%. The relevant descriptions have been added in Abstract (Line 29), Discussion (Line 355-356), and Conclusion (Line 525-526).

Reviewer 2 Report
Comments and Suggestions for Authors
The authors need to clarify the optimum dose, as both 0.3% and 0.6% show good outcomes.
The number of samples should be increased, as it is not acceptable to use 2 samples only.
Most of the fatty acids measured are short-chain fatty acids. Hence, use short-chain fatty acids rather than fatty acids, as it refers to long-chain fatty acids.
"All data were analyzed using the independent sample t-test in IBM SPSS Statistics software (version 25.0). Data analysis was conducted using one-way analysis of variance (ANOVA). The CON group (without FOS supplement in diet) was set as control." The sentence seems incorrect and needs to be corrected.
The microbiota analysis needs to be reanalyzed with an appropriate statistical analysis method.
Histopathology of investigated organs is recommended.
Author Response
We are grateful for your important suggestions and critical comments. In accordance with the reviewer concerns, we have revamped whole manuscript. Here are our point-to-point responses to your comments. Our responses are given directly in a different color (blue). The changes made when revising the manuscript are highlighted in a different color (red).
1.The authors need to clarify the optimum dose, as both 0.3% and 0.6% show good outcomes.
Response: Through this study, we determined that the optimal dosage is a 0.6% FOS supplementation. We have emphasized relevant descriptions in the text, such as: The 0.6% group exhibited the highest growth performance and cold carcass weight (Table 1); the 0.6% supplementation group exhibited the optimal enhancement of humoral immunity (lines 286–287); and so on. Based on the outcomes of this experiment, we conclude that 0.6% FOS is the appropriate concentration in feed for growing rabbits. The relevant descriptions have been added in Abstract (Line 29), Discussion (Line 355-356), and Conclusion (Line 525-526).
- The number of samples should be increased, as it is not acceptable to use 2 samples only.
Response: For each result, a minimum of three biological replicates were included. The number of samples corresponding to all results are described in the respective figure/table captions (Line 88, Line 110, Line 125, Line 131, Line 157, Line 166, Line 175, Line 190, Line 206, Line 208, Line 222, Line 226, Line 240, Line 557, Line 560, and Line 563).
- Most of the fatty acids measured are short-chain fatty acids. Hence, use short-chain fatty acids rather than fatty acids, as it refers to long-chain fatty acids.
Response: The relevant descriptions in the manuscript have been revised according to the reviewer's comments (Lines 167, 174, 479, and 561).
- "All data were analyzed using the independent sample t-test in IBM SPSS Statistics software (version 25.0). Data analysis was conducted using one-way analysis of variance (ANOVA). The CON group (without FOS supplement in diet) was set as control." The sentence seems incorrect and needs to be corrected.
Response: Thanks for your comments. The revised text is now presented in 4.10. Statistics (Lines 503-510).
- The microbiota analysis needs to be reanalyzed with an appropriate statistical analysis method.
Response: Thanks for your valuable suggestions. Non-parametric tests (Kruskal-Wallis test) was employed for significance analysis of differences. The relevant description has been added in 4.10. Statistics (Lines 503-510). We have supplemented the beta-diversity analysis (NMDS), as shown in Figure 6 (Line 207). NMDS analysis was performed on bacterial communities based on the Bray-Curtis distance algorithm. The stress value was 0.1370 (less than 0.2), indicating a good fit of the model (Line 195-197). The results of β-diversity analysis indicate that bacterial community structures of these groups were similar (Lines 197-199). Furthermore, after re-analyzing the α-diversity data using non-parametric tests, no significant differences were found among these groups (Line 193-194). These findings are consistent with our conclusion, suggesting that FOS does not disrupt the established gut microbiota equilibrium in rabbits.
- Histopathology of investigated organs is recommended.
Response: Throughout the experimental period, the rabbits remained healthy with no observed clinical signs of pathology (Line 424-426). Accordingly, histopathological analysis was not performed in the present study except HE slide of intesine. We have added a new figure (Figure 3) showing the HE-stained section of the intestine (Line 152). We sincerely appreciate your insightful suggestion, which has significantly enhanced our research perspective. In future investigations, we will employ disease-challenge models (such as pathogen or stress-induced enteritis) to evaluate the functional efficacy of FOS, complemented by histopathological examinations to holistically assess FOS mechanisms.

Reviewer 3 Report
Comments and Suggestions for Authors
This manuscript investigates the impact of dietary oligofructose (FOS) supplementation at different concentrations on growth, antioxidant capacity, immunity, gut morphology, and microbiota composition in growing rabbits. The study is well-structured, presents extensive experimental data, and holds practical relevance for animal production and gut health research. However, the following comments need to be addressed.
- While ANOVA and Tukey's test are mentioned, it is not always clear whether all assumptions for parametric tests (normality, variance homogeneity) were checked. For microbiota data, non-parametric tests might sometimes be more appropriate. Please clarify whether the data were checked for normality and equal variance before applying ANOVA. Indicate which variables (if any) required data transformation or non-parametric analysis.
- Multiple results are described as significant, but p-values are sometimes provided only as "<0.01" or "0.05". Some tables could benefit from explicit indication of n(number of animals per group/measurement). Please add sample sizes to all tables and figure legends. When possible, report exact p-values for key comparisons.
- The authors conclude that FOS does not "disrupt" microbiota but "maintains balance." However, the evidence for balance or health benefit based on sequencing is limited to changes in select taxa, alpha diversity, and family-level abundance. Consider adding a beta-diversity analysis such as PCoA or NMDS, and discuss whether overall community structure differs meaningfully between groups. Are shifts in specific taxa such as Lachnospiraceae and Monoglobus large enough to be linked to functional outcomes? Did the authors perform any correlation analysis between the abundance of specific taxa and physiological indices, such as enzyme activity, immunity markers, or growth performance? This could strengthen the proposed mechanisms.
- While the data show beneficial effects on many parameters, the mechanistic links between FOS, microbial shifts, immune enhancement, and growth are mostly correlative. Future studies might employ approaches such as fecal microbiota transplantation or antibiotic pretreatment to dissect the causality between microbiota changes and host effects.
- All animals used were healthy. The potential of FOS to prevent or treat intestinal pathology like enteritis is mentioned as a future direction; inclusion of a model system for gut disruption would further elevate the impact. Do the authors have data or plans for evaluating FOS efficacy in a disease-challenge model such as pathogen or stress-induced enteritis?
- The results suggest that 0.6% is optimal for multiple parameters, while 0.9% may have adverse or diminishing returns like lower acetic acid, changes in organ indices. Was any negative effect (clinical signs, poor growth, or organ pathology) observed in rabbits receiving 0.9% FOS? Please discuss potential mechanisms for the observed inhibitory effects at the highest dose and clearly state dietary recommendation boundaries for FOS use.
- Minor grammatical corrections are recommended throughout for clarity and conciseness. Please define all abbreviations upon first use in both the abstract and main text.
- Would prolonged FOS supplementation (beyond 72 days) continue to yield benefits, or might adaptation or negative effects emerge?
- The authors report improved meat characteristics; did they evaluate any sensory parameters or consumer acceptance?
- Are there functional predictions such as PICRUSt or shotgun metagenomics available for the changed microbiota composition, beyond taxonomic shifts?
- Have the authors considered whether similar doses or effects might be expected in other herbivorous or omnivorous livestock?
- Since the FOS used was extracted from a specific species, does its composition or degree of polymerization differ from commercial FOS, and could this influence the results?
Author Response
We are grateful for your important suggestions and critical comments. In accordance with the reviewer concerns, we have revamped whole manuscript. Here are our point-to-point responses to your comments. Our responses are given directly in a different color (blue). The changes made when revising the manuscript are highlighted in a different color (red).
- While ANOVA and Tukey's test are mentioned, it is not always clear whether all assumptions for parametric tests (normality, variance homogeneity) were checked. For microbiota data, non-parametric tests might sometimes be more appropriate. Please clarify whether the data were checked for normality and equal variance before applying ANOVA. Indicate which variables (if any) required data transformation or non-parametric analysis.
Response: We thank the reviewer for this valuable suggestion. All data were analyzed using IBM SPSS Statistics software (version 25.0). Data were first subjected to normality testing using the Shapiro-Wilk test. For datasets conforming to normal distribution, Bartlett's test was employed to assess homogeneity of variance. When homogeneity of variance was confirmed, one-way analysis of variance (ANOVA) was performed; otherwise (heterogeneous variance or non-normal distribution), the non-parametric Kruskal-Wallis test was applied. Unless otherwise stated, all data met the assumption of homogeneity of variance. Kruskal-Wallis test was employed for significance analysis of differences for 16S rDNA sequencing data. The relevant description has been added in 4.10 Statistics (Line 503-510).
- Multiple results are described as significant, but p-values are sometimes provided only as "<0.01" or "0.05". Some tables could benefit from explicit indication of n(number of animals per group/measurement). Please add sample sizes to all tables and figure legends. When possible, report exact p-values for key comparisons.
Response: Modified the p-value descriptions according to the comment. Explicit indication of n (number of animals per group/measurement) has been clearly stated in all figure and table captions (Line 88, Line 110, Line 125, Line 131, Line 157, Line 166, Line 175, Line 190, Line 206, Line 208, Line 222, Line 226, Line 240, Line 557, Line 560, and Line 563). Thank you for your suggestions, which have been very helpful in improving the quality of our manuscript.
- The authors conclude that FOS does not "disrupt" microbiota but "maintains balance." However, the evidence for balance or health benefit based on sequencing is limited to changes in select taxa, alpha diversity, and family-level abundance. Consider adding a beta-diversity analysis such as PCoA or NMDS, and discuss whether overall community structure differs meaningfully between groups. Are shifts in specific taxa such as Lachnospiraceae and Monoglobus large enough to be linked to functional outcomes? Did the authors perform any correlation analysis between the abundance of specific taxa and physiological indices, such as enzyme activity, immunity markers, or growth performance? This could strengthen the proposed mechanisms.
Response: Thanks for your valuable comments. We have supplemented the beta-diversity analysis (NMDS), as shown in Figure 6 (Line 207). NMDS analysis was performed on bacterial communities based on the Bray-Curtis distance algorithm. The stress value was 0.1370 (less than 0.2), indicating a good fit of the model (Line 195-197). The results of β-diversity analysis indicate that bacterial community structures of these groups were similar (Lines 197-199). Furthermore, after re-analyzing the α-diversity data using non-parametric tests, no significant differences were found among these groups (Line 193-194). These findings are consistent with our conclusion, suggesting that FOS does not disrupt the established gut microbiota equilibrium in rabbits. However, due to limitations in resources and funding at the time, we did not perform correlation analysis between the abundance of specific taxa and physiological indices. Although alterations in the abundance of Lachnospiraceae and Monoglobus were observed, their associations with functional outcomes require further investigation to confirm. We sincerely appreciate your suggestion, which has been highly insightful for our research. In future studies, we will conduct experiments evaluating the efficacy of FOS in a disease-challenge model (such as stress-induced enteritis) to further elucidate the proposed mechanisms. The relevant description has been added in Line 336-340.
- While the data show beneficial effects on many parameters, the mechanistic links between FOS, microbial shifts, immune enhancement, and growth are mostly correlative. Future studies might employ approaches such as fecal microbiota transplantation or antibiotic pretreatment to dissect the causality between microbiota changes and host effects.
Response: We sincerely appreciate the reviewer's insightful suggestion regarding the need to establish causal relationships between FOS-induced microbial shifts and host physiological outcomes. We fully acknowledge that the current findings primarily demonstrate correlative relationships rather than mechanistic causality. As such, we agree that approaches such as fecal microbiota transplantation (FMT) and targeted antibiotic depletion would be invaluable for disentangling these complex interactions. We believe these approaches will significantly advance our understanding of the causal links between FOS supplementation, gut microbiome remodeling and host immunity responses. The relevant description is supplemented in Line 336-340.
- All animals used were healthy. The potential of FOS to prevent or treat intestinal pathology like enteritis is mentioned as a future direction; inclusion of a model system for gut disruption would further elevate the impact. Do the authors have data or plans for evaluating FOS efficacy in a disease-challenge model such as pathogen or stress-induced enteritis?
Response: Thank you for your suggestions, which are very helpful to us. In future studies, we will evaluate the efficacy of FOS in disease-challenge models (such as stress-induced enteritis) to further elucidate the proposed mechanisms. The relevant description is supplemented in Line 336-340.
- The results suggest that 0.6% is optimal for multiple parameters, while 0.9% may have adverse or diminishing returns like lower acetic acid, changes in organ indices. Was any negative effect (clinical signs, poor growth, or organ pathology) observed in rabbits receiving 0.9% FOS? Please discuss potential mechanisms for the observed inhibitory effects at the highest dose and clearly state dietary recommendation boundaries for FOS use.
Response: During the entire experimental process, we did not observe any negative effects (such as clinical signs, poor growth, or organ pathology) in rabbits among these groups (including the FOS-3 group). All rabbits remained healthy with no observed clinical signs of pathology (Line 411-413). While FOS improves resilience to stressors (e.g., weaning, disease), excessive supple-mentation may paradoxically impair antioxidant function in previous study (Line 276-279). Thus, optimizing FOS dosage is critical to balancing its benefits in lipid regulation, antioxidant enhancement and stress resistance. In this study, the highest dose of FOS (0.9%) showed potential adverse effects on cecal fermentation and organ indices, but no negative effects in clinical signs, poor growth, or organ pathology were observed. Higher FOS supplementation would also increase costs. Therefore, we recommend that the addition level in meat rabbits should not exceed 0.9%. The recommended dose is 0.6%. The relevant description is supplemented in Line 351-356.
- Minor grammatical corrections are recommended throughout for clarity and conciseness. Please define all abbreviations upon first use in both the abstract and main text.
Response: Thank you for your critical comments. The whole manuscript has been polished according to the comment.
- Would prolonged FOS supplementation (beyond 72 days) continue to yield benefits, or might adaptation or negative effects emerge?
Response: We appreciate the reviewer's insightful question regarding prolonged FOS supplementation. Our study demonstrated that during the growth period (34–106 days), 0.6% FOS supplementation consistently enhanced immune function, antioxidant capacity, and gut health in rabbits. However, we acknowledge the need to explore effects beyond this timeframe. We value this suggestion highly and will extend FOS administration to ≥100 days in future investigations to determine whether adaptation or negative effects would emerge. These works will establish guidelines for prolonged FOS use in commercial rabbit production.
- The authors report improved meat characteristics; did they evaluate any sensory parameters or consumer acceptance?
Response: Due to the scope of our initial experimental design, sensory parameters and consumer acceptance were not evaluated. We sincerely appreciate the reviewer's valuable suggestions, which will significantly improve our future research protocols. In subsequent studies, we will systematically incorporate sensory analyses. We will also conduct consumer acceptance evaluations if possible.
- Are there functional predictions such as PICRUSt or shotgun metagenomics available for the changed microbiota composition, beyond taxonomic shifts?
Response: Due to limitations in resources and funding during this study, functional predictions were not performed. However, we fully acknowledge that methodologies such as PICRUSt or shotgun metagenomics are essential for establishing mechanistic links between gut microbiota and key parameters including growth performance and immune function. We sincerely appreciate the reviewer's suggestion and will implement these advanced microbiome analysis approaches in ongoing investigations to further elucidate the proposed biological mechanisms.
- Have the authors considered whether similar doses or effects might be expected in other herbivorous or omnivorous livestock?
Response: We appreciate the reviewer's insightful question regarding cross-species extrapolation. While our current study focuses on rabbits, similar investigations have been conducted in other herbivorous and omnivorous species such as rats, dairy cows and other livestock (as referenced in Lines 263-264, 272-274, 322-323 and so on). These findings support the broader applicability of FOS in diverse livestock.
We recognize the importance of this question and will expand our research direction to include larger livestock species such as pigs, cattle or sheep to further explore the potential of FOS across a wider range of species. This extension will help address the broader applicability of our findings and contribute to the broader field of nutritional research.
- Since the FOS used was extracted from a specific species, does its composition or degree of polymerization differ from commercial FOS, and could this influence the results?
Response: FOS was supplied by the National Engineering Research Center of Juncao Technology (Lines 386–387), which maintains mature extraction and processing protocols yielding >95% purity (HPLC-verified). Therefore, we believe that its composition and degree of polymerization are comparable to commercial FOS standards, and could not influence the results.

Round 2
Reviewer 2 Report
Comments and Suggestions for Authors
The authors followed all the comments.
Reviewer 3 Report
Comments and Suggestions for Authors
No further comments